# Model ensemble instead of prompt fusion: a sample-specific knowledge transfer method for few-shot prompt tuning

**Xiangyu Peng**[†] **Chen Xing**[‡] **Prafulla Kumar Choubey**[‡] **Chien-Sheng Wu**[‡] **Caiming Xiong**[‡]
[†]Georgia Institute of Technology          [‡]Salesforce Research
xpeng62@gatech.edu, {cxing,pchoubey,wu.json,cxiong}@salesforce.com

## Abstract

Prompt tuning approaches, which learn task-specific soft prompts for a down-stream task conditioning on frozen pre-trained models, have attracted growing interest due to its parameter efficiency. With large language models and sufficient training data, prompt tuning performs comparably to full-model tuning. However, with limited training samples in few-shot settings, prompt tuning fails to match the performance of full-model fine-tuning. In this work, we focus on improving the few-shot performance of prompt tuning by transferring knowledge from soft prompts of source tasks. Recognizing the good generalization capabilities of ensemble methods in low-data regime, we first experiment and show that a simple ensemble of model predictions based on different source prompts, outperforms existing multi-prompt knowledge transfer approaches such as source prompt fusion in the few-shot setting. Motivated by this observation, we further investigate model ensembles and propose **S**ample-specific **E**nsemble of **So**urce **M**odels (SESoM). SESoM learns to adjust the contribution of each source model for each target sample separately when ensembling source model outputs. Through this way, SESoM inherits the superior generalization of model ensemble approaches and simultaneously captures the sample-specific competence of each source prompt. We conduct experiments across a diverse set of eight NLP tasks using models of different scales (T5-{base, large, XL}) and find that SESoM consistently outperforms the existing models of the same as well as larger parametric scale by a large margin.

## 1 Introduction

Recent few years have witnessed the great success of large pre-trained language models (PLM) (Kenton & Toutanova, 2019; Liu et al., 2019; Radford et al., 2019; Raffel et al., 2020; Brown et al., 2020). The size of pre-trained models which can easily go to billions of parameters (Brown et al., 2020; Raffel et al., 2020), however, hinder their real-world deployments and applications. The huge size of pre-trained language models can make model fine-tuning for downstream NLP tasks computationally expensive and memory-inefficient. To alleviate this problem, many parameter-efficient fine-tuning methods are proposed (Li & Liang, 2021; Houlsby et al., 2019; Zhang et al., 2021; Lester et al., 2021; Liu et al., 2021b). Among them, *prompt tuning* (Lester et al., 2021) is one of the most widely adopted methods. Given a downstream task, prompt tuning methods keep the entire pre-trained model frozen. Only the newly added task-specific *soft prompts* are updated on the training data from a target task, conditioning on the original pre-trained model. Compared to traditional fine-tuning methods that update the entire pre-trained model, prompt tuning consumes significantly less memory and less training time per iteration (Table 10 in Gu et al. (2022)).

Despite prompt tuning's advantages in practice and its continuously improved performances on various NLP tasks (Liu et al., 2021a; Vu et al., 2022), its performances in few-shot settings where labeled training data is limited, still have large space for improvements (Gu et al., 2022). In low-data scenarios, one of the most widely applied approaches to alleviate data shortage of the *target task*, is to seek help from *source tasks* where labeled training data is abundant. Although such knowledge transfer approaches from multiple source tasks are analyzed on full-model training in other domains

(Chen, 2021; Li et al., 2020; Sasso et al., 2022; Lee et al., 2019), relevant methods for few-shot prompt-tuning still remain under explored. Therefore, in this work, we seek to find an effective strategy to use trained soft prompts from multiple source tasks to benefit few-shot prompt tuning on a new target task.

With soft prompts trained from several source tasks and full training data from a target task, there are a few existing approaches one could adopt. Vu et al. (2022) finds the most suitable source soft prompt to initialize the soft prompt of the target task. Alternatively, Asai et al. (2022) directly fuses all source soft prompts together with a target task-specific prompt. Although both source soft prompt based initialization and fusion improve performance with enough training data for a target task, we empirically find them not as effective under few-shot settings. Another tempting alternative we could employ to use source prompts is model ensemble, which is known to provide good generalization and low variance (Hansen & Salamon, 1990). For instance, Dvornik et al. (2019) and Liu et al. (2020) show that simple ensemble methods outperform complicated approaches in few-shot settings in the computer vision domain. Therefore, for few-shot prompt tuning, we wonder whether an ensemble of model outputs given different source prompts achieve better performance compared to existing approaches employing source prompts. If so, what is the most effective model ensemble strategy for the knowledge transfer from multiple source prompts?

To answer these questions, we conduct empirical analysis and find that a simple uniform logit-averaging ensemble of model predictions based on different source prompts, can already outperform existing multi-source knowledge transfer approaches for few-shot prompt tuning. Motivated by this observation, we further look into ensemble approaches and propose our solution, a sample-specific ensemble of source models (SESoM). Source models refer to the trained soft prompts of source tasks, together with the pre-trained language model that the source soft prompts are trained with. As the name suggests, SESoM learns from the few-shot target samples to adaptively decide how much each source task should contribute given different target samples. Specifically, our method trains an attention-style network to generate weights to ensemble the outputs of different source models, in order to make the prediction given each target sample. Through this way, our model is able to capture the sample-specific preferences to ensemble different source models given the few-shot labelled target data. Therefore, compared to existing knowledge transfer approaches for prompt tuning that provide a fixed knowledge transfer strategy for all target samples, SESoM is more effective due to its sample-specific strategy.

We conduct experiments across six source tasks and eight target tasks on three model scales, T5-Base, T5-Large and T5-XL. Experimental results show that SESoM outperforms existing methods, such as source prompt fusion approaches and other model ensemble methods, by a large margins in every scenario tested. Moreover, we also find that SESoM can consistently achieve better performance compared to existing methods when the number of few-shot labeled target data increases. Even in full-data settings, SESoM outperforms existing methods although not as significantly as in few-shot settings. Finally, we find that SESoM can achieve better performance when the number of source tasks increases, even when the newly added tasks are less preferable in general for the target task. Our case study also shows that SESoM can generate different ensemble weights for different samples of one target task. The generated weights are also aligned with the sample-specific performance of different source models.

## 2 RELATED WORK

**Knowledge transfer approaches in the context of prompt tuning.** Since the emergence of prompt tuning methods, much recent research has focused on improving the performance of prompt-tuning methods on full-data fine-tuning. Some of them focus on transferring knowledge from other tasks which are similar to the target task, to facilitate prompt tuning of the target task. Among them, SPoT (Vu et al., 2022) first learns a prompt on one or more source tasks and then uses it to initialize the prompt for a target task. SPoT significantly boosts the performance of prompt-tuning across many tasks. Similarly, PPT (Gu et al., 2022) pre-trains the soft prompt of the target task with data formulated similarly with target data. These two methods provide a fixed knowledge transfer strategy for all target samples, given that they both provide initialization before few-shot prompt tuning of the target task. Different from them, our method provides sample-specific knowledge transfer from source models to each target samples, leading to better performance on few-shot fine-

tuning. More recently, Asai et al. (2022) proposes a sample-specific prompt fusion method for full-data prompt tuning. In such method, the soft prompts of source tasks are fused together to construct a new prompt for each target sample, instead of ensembling the source models' outputs given each target sample in SESoM.

**Ensemble learning** has been a popular approach to obtain a low-variance and generalizable model (Hansen & Salamon, 1990). The basic ensemble techniques include voting (Hansen & Salamon, 1990), bagging (Breiman, 1994), boosting (Schapire, 1990; Freund & Schapire, 1997) and stacking (Wolpert, 1992), which have been applied across many NLP problems such as model debiasing (Elazar & Goldberg, 2018; Stacey et al., 2020), cross-lingual transfer learning (Wang et al., 2021), calibrating sequence classification and generation models (Reich et al., 2020), domain adaptation (Kim et al., 2017). Within the paradigm of prompt-based learning, Schick & Schütze (2021a) explores unweighted average of logits corresponding to different human-selected verbalizers for zero-shot classification. Lester et al. (2021) uses majority voting over logits from five prompts trained on the same task. They all employ identical ensemble strategy for all test samples, which could be sub-optimal. Schick & Schütze (2021a) also found that the oracle that selects the best verbalizer for every test sample significantly outperforms the unweighted-averaging model. SESoM fills this gap by taking over the role of oracle and train a separate attention module to generate sample-specific importance weight for every source model. The sample-specific weights help to measure each source model's competence in making prediction for a sample, and enable better utilization of available prompts.

## 3 METHOD

### 3.1 PRELIMINARIES

In SESoM, given the training data for source tasks $S_1, ..., S_T$ and a pre-trained language model, we first train a soft prompt $\mathbf{P}_j$ ($j \in [1, T]$) for each source task by running prompt tuning (Lester et al., 2021). Following a typical T5-setting, we restructure all downstream tasks to text-to-text generation format, where each label of a training sample is represented by a *verbalizer* (Schick & Schütze, 2021c) and, optionally, a task-specific *template* (Schick & Schütze, 2021c;b; Mishra et al., 2021). We represent an instance in a source or target task as $(\mathbf{X}, y)$, where $\mathbf{X}$ is a sequence of token embeddings ($\mathbf{X} = [\mathbf{x}_1, ..., \mathbf{x}_l] \in \mathbb{R}^{l \times d}$, $l$ is the length of the input token sequence and $d$ is the embedding size of PLM), and $y$ is a classification label. Then, we map the class label $y$ to its corresponding verbalizer or the verbalizer-template sequence, and call it $Y^1$. Each soft prompt $\mathbf{P}_j = [\mathbf{p}_1, ..., \mathbf{p}_m] \in \mathbb{R}^{m \times d}$ is also a sequence of embeddings, where $m$ is the number of soft prompt embeddings for the task.

Prompt tuning prepends a randomly initialized task-specific soft prompt $\mathbf{P}_j$ to the input embedding sequence $\mathbf{X}$, resulting in the final input $[\mathbf{P}_j; \mathbf{X}]$. Then, $[\mathbf{P}_j; \mathbf{X}]$ is fed into the pre-trained model to make the prediction. The target task is modeled as $\text{Pr}_\theta(\mathbf{Y} \mid [\mathbf{P}_j; \mathbf{X}])$ where the parameters of the pre-trained model ($\theta$) is frozen during the tuning and only the soft prompt $\mathbf{P}_j$ is updated in order to maximize $\text{Pr}_\theta(\mathbf{Y} \mid [\mathbf{P}_j; \mathbf{X}])$. We show the process of prompt tuning in Fig.1 (a).

### 3.2 SAMPLE-SPECIFIC ENSEMBLE OF SOURCE MODELS

Given a collection of source prompts $[\mathbf{P}_1; ...; \mathbf{P}_T]$ from source tasks $[S_1; ...; S_T]$ trained using prompt-tuning, and the pre-trained language model $\theta$ that these soft prompts are conditioned on, SESoM aims to use their knowledge on a new target task under few-shot settings. We call the prompt from a source task together with the PLM as a source model, represented as $[\mathbf{P}_j; \theta]$. In SESoM, we first train each source model $[\mathbf{P}_j; \theta]$ with the few-shot target data in a prompt-tuning manner. This enforces the source models to generate target task's verbalizers given the target input sample.

Then taking a labeled instance $(\mathbf{X}, y)$ from the few-shot target task $T_{\text{target}}$, SESoM first feeds $[\mathbf{P}_j; \mathbf{X}]$ into the corresponding source model $[\mathbf{P}_j; \theta]$ and obtain the pre-softmax logit $\mathbf{l}_{x,j}$. It then uses the $\mathbf{l}_{x,j}$ and $X$ to compute sample-specific attention weight representing the competence of the

---

[1]We detail all the verbalizers used in our experiments in Appx. B.5.

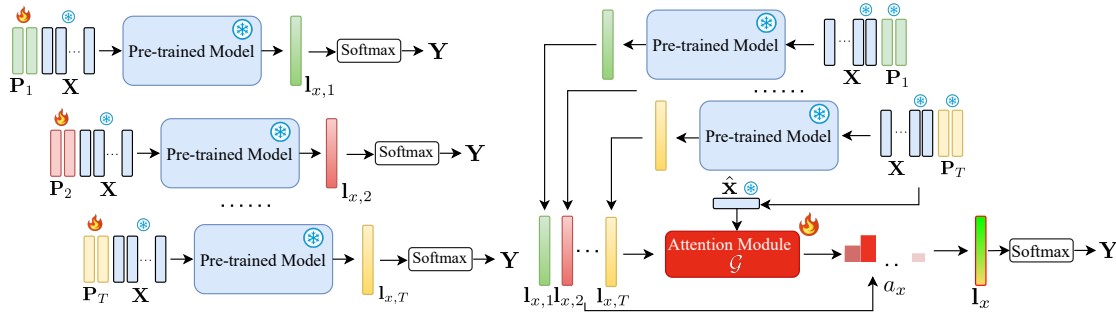

(a) Training of source prompts.      (b) Training of attention module $\mathcal{G}$.

Figure 1: Overview of SESoM. Components with 🔥 are updated during training while those with ❄ are frozen. (a) describes the training of source prompts in SESoM. Given the input token embedding sequence $\mathbf{X}$ of different source tasks and a pre-trained model, prompt tuning trains task specific source prompts $[\mathbf{P}_1; ...; \mathbf{P}_T]$ for source tasks $[S_1; ...; S_T]$ by prepending them to $\mathbf{X}$ as inputs and feeding to the frozen pre-trained model. (b) describes how SESoM trains the attention module $\mathcal{G}$. $\mathcal{G}$ takes the logits $[\mathbf{l}_{x,1}; ...; \mathbf{l}_{x,T}]$ from all $T$ source models as input and uses $\hat{\mathbf{x}}$ as the key to calculate the attention weights $\mathbf{a}_x$ of source logits. Then source logits are weighted averaged accordingly to construct $\mathbf{l}_x$ for the final prediction.

source model $[\mathbf{P}_j; \theta]$ for the given input $X$. We train an attention-style network $\mathcal{G}$ (§ 3.2.1) to compute these sample-specific weights for every source model. Finally, it takes the weighted average of logits ($\mathbf{L}_x = [\mathbf{l}_{x,1}; ...; \mathbf{l}_{x,T}] \in \mathbb{R}^{T \times v}$, where $v$ is the vocabulary size of the pre-trained model) from all source models using their attention weights to make its prediction for $\mathbf{X}$.

### 3.2.1 ATTENTION MODULE $\mathcal{G}$.

The goal of the attention module $\mathcal{G}$ is to learn to generate weights to ensemble the source models' pre-softmax logits based on their competence on a target sample $X$. One input of $\mathcal{G}$ is therefore the representation of target sample $X$. Other candidate inputs of $\mathcal{G}$ which might benefit this sample-specific weight generation process are either source prompts or pre-softmax logits of source models. We empirically find using $X$ and source prompts as input for $\mathcal{G}$ are less effective compared to taking $X$ and source model logits as $\mathcal{G}$ inputs under few-shot settings. We hypothesize that it is because prompts are not self-sufficient as their competency is not solely determined by them. They are effective only when used along with the PLM for which it was tuned. With abundant training samples, the prompt-based attention module may learn to identify source prompts that are relevant for an input sample from the target task. But with few training samples, it is unable to learn to accurately weight predictions from different source models, required to capture their effectiveness for every sample. Logits of source models, on the other hand, are a function of both the prompt and the PLM and are also capable of modeling the uncertainty in predictions from different models. Consequently, we propose to use the source model logits $\mathbf{l}_{x,j}$ along with the input $X$ in our attention module.

We first perform max-pool over the token embedding sequence $\mathbf{X} = [\mathbf{x}_1; ...; \mathbf{x}_l] \in \mathbb{R}^{l \times d}$, to obtain $\hat{\mathbf{x}} \in \mathbb{R}^d$ as the representation of $X$, Then given the representation $\hat{\mathbf{x}}$ of the target input $X$, we apply a feed-forward network to project $\hat{\mathbf{x}}$ non-linearly to a new representational space,

$$\mathbf{h}_x = W_{u,x}^T \cdot \gamma(W_{d,x}^T \cdot \hat{\mathbf{x}}) \tag{1}$$

where $\gamma(\cdot)$ is a non-linear activation function, $W_{d,x} \in \mathbb{R}^{d \times d'_x}$ and $W_{u,x} \in \mathbb{R}^{d'_x \times d'}$ are trainable weights. Then we apply Layer Norm (Ba et al., 2016) to get $\mathbf{h}_x \in \mathbb{R}^{d'}$ — the final projected representation of target input $X$. Similarly, we project the pre-softmax logit $\mathbf{l}_{x,j} \in \mathbb{R}^v$ of each source model given $\mathbf{X}$ into the same space,

$$\mathbf{h}_{l,j} = W_{u,l}^T \cdot \gamma(W_{d,l}^T \cdot \mathbf{l}_{x,j}), \tag{2}$$

where $W_{d,l} \in \mathbb{R}^{v \times d'_l}$ and $W_{u,l} \in \mathbb{R}^{d'_l \times d'}$ are trainable weights. Then, given $\mathbf{h}_x$ and the projected representations of all source logits $\{\mathbf{h}_{l,j}\}_{j=1}^T$, we compute the attention score $a_{x,j}$ for the source

logit $\mathbf{l}_j$ by

$$a_{x,j} = \frac{e^{(\mathbf{h}_{l,j} \cdot \mathbf{h}_x)}}{\sum_{k=1}^{T} e^{(\mathbf{h}_{l,k} \cdot \mathbf{h}_x)}}. \tag{3}$$

SESoM produces the final output logit $\mathbf{l}_x \in \mathbb{R}^v$ by computing a linear combination of $[\mathbf{l}_{x,1}; ...; \mathbf{l}_{x,T}]$ given the computed input-logit attention,

$$\mathbf{l}_x = \mathcal{G}(\hat{\mathbf{x}}, [\mathbf{l}_{x,1}; ...; \mathbf{l}_{x,T}]) = \sum_{j=1}^{T} a_{x,j} \mathbf{l}_{x,j} \tag{4}$$

Then following the prompt tuning approach in Section 3.1, we freeze all the parameters $\theta$ in pre-trained language model and source soft prompts ($[\mathbf{P}_1; ...; \mathbf{P}_t]$). We also show the forward pass of SESoM in Fig.1 (b). Attention module $\mathcal{G}$ is trained by minimizing its cross-entropy loss between softmax($\mathbf{l}_x$) and the label $\mathbf{Y}$. During the few-shot training, we update the attention module $\mathcal{G}$ with the few-shot labeled target samples. Through this way, $\mathcal{G}$ is trained to capture the sample-specific preference of different source models. At inference, we also use $\mathcal{G}$ to calculate the sample specific ensemble weight of all source logits, and calculate the weighted average of them as the final logit to make the prediction.

**Parameter efficiency.** As mentioned in Lester et al. (2021), prompt tuning methods are a natural fit for model ensemble approaches from the perspective of parameter efficiency. Unlike other models for which an ensemble of $T$ models leads to $T$ times more model parameters, an ensemble of $T$ different prompt-tuning models only leads to $T$ times more soft prompts. Because the pre-trained model that soft prompts are trained to condition on, is identical for all models to be ensembled. Therefore, although SESoM is a model ensemble approach, the additional model parameters introduced by the ensemble are only the soft prompts of 6 source tasks, i.e., $T \times m \times d$ parameters (0.6M in our experiment). SESoM also trains one attention module which includes four projection layers and two layer norms. It requires $d \times d'_x + d'_x \times d' + v \times d'_l + d'_l \times d' + 4d'$ parameters ($\sim 0.9$M in our experiment). Therefore, the total number of additional trainable model parameters in SESoM is only less than $0.5\%$ of a pre-trained T5-base model.

## 4 EXPERIMENTS

In this section, we first describe our experimental setup, then our main empirical results compared to baselines. Finally we conduct various empirical analysis and case study to show the effectiveness of different components of SESoM and reasonings behind our design choices.

### 4.1 EXPERIMENTAL SETUP

**Few-shot prompt tuning setups and hyper-parameters of SESoM.** For our main experiments, we follow existing work (Gu et al., 2022) to set the number of few-shot training and validation samples for every target task as 32. We evaluate all models with the full original validation set of every target task. All presented results are an average of 20 runs with different random seeds. For SESoM, we randomly initialize attention module $\mathcal{G}$ based on Section 3.2. Details of other hyper-parameters of SESoM, such as the embedding sizes of $\mathcal{G}$, the optimizer and learning rate during training, etc, can be found in Appendix B.3. We use pre-trained T5-{base,large,XL}(Raffel et al., 2020) as language model $\theta$.

**Source and target tasks.** Following existing work(Asai et al., 2022), we use six tasks that have been considered generally helpful to other NLP tasks as our source tasks. They are MNLI (Williams et al., 2018), QNLI (Demszky et al., 2018), QQP (Wang et al., 2018), SST2 (Socher et al., 2013), SQuAD (Rajpurkar et al., 2016), and ReCoRD (Zhang et al., 2018). Given each pre-trained model tested (T5-{base,large,XL}), we train soft prompts for these source tasks separately using prompt tuning. The hyper-parameters for training the soft prompts can be found in Appendix B.1.

We then evaluate our model on the following GLUE (Wang et al., 2018) and SuperGLUE (Wang et al., 2019) tasks: WNLI (Wang et al., 2018), MRPC (Dolan et al., 2005), BoolQ (Clark et al., 2019), MultiRC (Khashabi et al., 2018), RTE (Giampiccolo et al., 2007), WiC (Pilehvar & Camacho-Collados, 2019), WSC (Levesque et al., 2012), and CB (De Marneffe et al., 2019). They cover a

Table 1: Few-shot performance of all methods. All the scores are the average of 20 runs with different random seeds, with standard errors included in subscripts. The best scores are in bold. "Uni Ensemble" is the acronym for Uniform Ensemble. "MV Ensemble" stands for Majority-vote Ensemble. "FW Ensemble" stands for Fixed-weight Ensemble.

| Model | Method | WNLI | MRPC | RTE | MultiRC | BoolQ | WiC | WSC | CB | Avg$_{std}$ |
|---|---|---|---|---|---|---|---|---|---|---|
| | SPoT-t | 54.44 | 53.32 | 54.15 | 50.61 | 56.87 | 54.51 | 47.98 | 43.48 | $51.92_{10.64}$ |
| | PPT | 47.89 | 68.97 | **67.51** | 60.32 | 51.81 | 57.25 | **63.46** | 50.00 | $58.40_{1.03}$ |
| | ATTEMPT | 52.95 | 75.35 | 52.63 | 59.52 | 54.28 | 54.48 | 43.89 | 43.74 | $54.61_{6.48}$ |
| T5-Base | Uni Ensemble | **56.34** | 75.00 | 57.40 | 65.92 | 70.12 | 62.69 | 45.19 | 57.14 | $61.22_{3.87}$ |
| | MV Ensemble | 50.24 | 74.65 | 59.87 | **71.39** | 73.97 | **71.59** | 49.71 | 65.82 | $64.66_{2.15}$ |
| | FW Ensemble | 50.07 | 77.57 | 65.03 | 66.32 | 72.28 | 63.14 | 51.39 | 43.74 | $62.28_{4.58}$ |
| | SESoM | **56.34** | **84.34** | 66.66 | 66.91 | **74.80** | 63.68 | 53.56 | **74.02** | **$67.54_{2.16}$** |
| | SPoT-t | 54.93 | 72.46 | 65.14 | 64.83 | 53.66 | 56.26 | 48.76 | 53.70 | $58.72_{14.23}$ |
| | PPT | 43.66 | 79.41 | 75.61 | 65.50 | 67.68 | 53.61 | 61.54 | 80.36 | $65.95_{1.00}$ |
| | ATTEMPT | 42.25 | 67.96 | 62.63 | 70.11 | 68.42 | 54.33 | 60.04 | 69.64 | $61.92_{6.37}$ |
| T5-Large | Uni Ensemble | **56.34** | 86.76 | 67.14 | 74.36 | 78.83 | 64.42 | 60.42 | 58.78 | $68.38_{3.35}$ |
| | MV Ensemble | 49.58 | 84.20 | 58.19 | 73.60 | 78.58 | **72.67** | 53.09 | 56.60 | $65.81_{2.88}$ |
| | FW Ensemble | **56.34** | 83.74 | 77.97 | 73.63 | **81.28** | 61.44 | 56.73 | 62.85 | $69.24_{3.19}$ |
| | SESoM | **56.34** | **87.84** | **82.80** | **74.40** | 81.24 | 67.06 | **63.08** | **84.73** | **$74.69_{0.89}$** |
| | SPoT-t | 56.34 | 77.65 | 75.16 | 51.07 | 60.97 | 59.53 | 50.58 | 70.89 | $62.77_{14.52}$ |
| | PPT | 50.07 | 76.22 | 75.83 | 67.57 | 69.89 | 58.62 | 59.73 | 78.92 | $67.10_{0.83}$ |
| | ATTEMPT | 49.37 | 62.46 | 84.24 | 71.26 | 60.39 | 42.50 | 45.38 | 80.18 | $61.97_{6.65}$ |
| T5-XL | Uni Ensemble | 56.34 | 75.73 | 89.16 | 76.56 | 81.07 | 58.77 | 63.40 | 78.57 | $72.45_{3.61}$ |
| | MV Ensemble | 41.29 | 73.46 | 84.24 | 75.30 | 77.48 | 64.43 | 49.49 | 82.83 | $68.57_{2.87}$ |
| | FW Ensemble | 56.27 | 85.62 | 89.58 | 76.20 | **82.83** | 66.79 | 57.55 | 73.48 | $73.54_{2.41}$ |
| | SESoM | **56.34** | **88.21** | **89.71** | **76.90** | 82.50 | **68.56** | **63.41** | **84.15** | **$76.22_{1.44}$** |

wide range of NLP tasks including question answering, sentiment analysis and textual entailment. Details of source and target tasks can be found in Appendix A.

**Baselines.** We compare SESoM not only with existing knowledge transfer approaches in the context of prompt tuning, but also with multiple existing ensemble strategies. The knowledge transfer methods for prompt tuning are,

- **SPoT-t** (Vu et al., 2022): SPoT-t originally first trains soft prompt with the full training set of the target task for a few epochs, and then retrieve the source soft prompt which shares the highest cosine similarity with this target prompt to re-initialize the target prompt. Then, it re-trains the target prompt starting from this new initialization. To adapt SPoT-t to few-shot settings, we keep everything unchanged except that we train the target prompt with the few-shot labelled data, instead of with the full target training set.
- **PPT**(Gu et al., 2022): PPT pre-trains the target prompt with additional data by pre-processing the additional data to the format of the target task. For fair comparison with other approaches, we use the training data of all 6 source tasks to pre-train the target soft prompt. We then update the target soft prompt with the few-shot target data as in prompt tuning.
- **ATTEMPT**(Asai et al., 2022): ATTEMPT is a sample-specific knowledge fusion method for full-data prompt tuning. In this method, the soft prompts of source tasks are fused together to construct a new prompt for each target sample, unlike SESoM's model-level ensemble of source models. To apply ATTEMPT in few-shot settings, we keep everything unchanged for ATTEMPT except that we train their model with the few-shot labelled data, instead of with the full target training set.

The ensemble baselines are,

- **Uniform Ensemble**: the simplest ensemble method of the source models. Same as SESoM, we first fine-tune each source model with the target few-shot labelled data. Then we take the average of pre-softmax logits of all source model's output given a target sample to make the prediction. The difference of Uniform-Ensemble from SESoM is that Uniform-Ensemble simply averages the source logits for all target samples instead of conducting a sample-specific weighted average.
- **Majority-Vote Ensemble**: in this ensemble baseline, we use every individual source model to make predictions first. Then we take the prediction that have the most votes as the final prediction. The rest setting of Majority-Vote Ensmeble is the same as Uniform Ensemble.

Table 2: SESoM with top k source models. All results are the average of 20 runs with different random seeds. Standard errors are included in subscripts. "# s." stands for the number of source models used in SESoM.

| Model | # s. | WNLI | MRPC | RTE | MultiRC | BoolQ | WiC | WSC | CB | $\text{Avg}_{\text{std}}$ |
|---|---|---|---|---|---|---|---|---|---|---|
| T5-Base | 1 | 54.44 | 53.32 | 54.15 | 50.61 | 56.87 | 54.51 | 47.98 | 43.48 | $51.92_{10.64}$ |
| | 3 | 56.97 | 72.59 | 62.49 | 61.59 | 67.48 | 61.43 | 48.08 | 64.11 | $61.84_{7.48}$ |
| | 5 | 56.41 | 81.91 | 64.01 | 66.08 | 72.96 | 61.54 | 49.04 | 69.46 | $65.18_{3.40}$ |
| T5-Large | 1 | 54.93 | 72.46 | 65.14 | 64.83 | 53.66 | 56.26 | 48.76 | 53.70 | $58.72_{14.23}$ |
| | 3 | 56.34 | 86.27 | 72.17 | 73.55 | 76.36 | 62.24 | 56.30 | 68.04 | $68.91_{6.93}$ |
| | 5 | 56.34 | 87.43 | 77.60 | 74.09 | 80.28 | 66.05 | 60.19 | 80.98 | $72.87_{3.41}$ |

- **Fixed-weight Ensemble**: Instead of simply averaging logits of source models as Uniform-Ensemble, Fixed-weight Ensemble takes a weighted average of the source logits to make predictions. The voting weight of each source model is fixed for all target samples. The voting weight is calculated according to the F1 score of the source models given the few-shot target data. The better a source model performs given the labelled few-shot data, larger voting weight it would have for all target samples.

More details about the training of all baseline methods above can be found in Appendix B.2.

## 4.2 MAIN RESULTS

The few-shot performance of SESoM and other methods are shown in Table 1. First, on every pre-trained model (T5-{base,large,XL}), SESoM outperforms all baselines on the average score by a large margin. SESoM also outperforms baselines on a definite majority of single target tasks on pre-trained models with different sizes. Specifically, compared to approaches that provide a fixed knowledge transfer strategy for all target samples (SPoT-t, PPT, Uni Ensemble, FW Ensemble), SESOM's superior performance indicates the effectiveness of its sample-specific strategy. Compared to existing approaches that conduct sample-specific soft prompt fusion (ATTEMPT), SESoM's superior performance indicates the effectiveness of its model ensemble strategy in few-shot settings. Moreover, if we compare SESoM on T5-Base with non-ensemble baselines (SPoT-t, PPT, ATTEMPT) on T5-Large, we can see that SESoM on T5-Base even outperforms these baselines with a larger pre-trained model. Similarly, SESoM on T5-Large also outperforms non-ensemble baselines on T5-XL. This indicates that SESoM further improves the parameter efficiency of prompt tuning in few-shot settings.

Second, if we compare the performances of ensemble methods (Uni Ensemble, FW Ensemble and our proposed method SESoM) versus the rest methods, ensemble methods have superior performance against the rest methods in the few-shot setting. The bigger the pre-trained model is, more apparent the performance advantages of ensemble methods are in the few-shot setting. It is likely due to the widely observed strengths of ensemble methods that provide a better generalization and lower variance during inference, as illustrated in Section 1 as our motivation. Finally, compared with all methods, ATTEMPT seems to have the most unsatisfactory results in few-shot settings. AT-TEMPT uses the few-shot target data to train a network to output weights to fuse source soft prompts for each target sample. Although it achieves great performances in full-data fine-tuning, few-shot training is not an easy scenario for it due to the overfitting caused by limited training data. While our proposed SESoM takes the logits of source models as input and outputs weights for model ensemble during few-shot training, which takes advantage of the superior generalization and robustness of model ensemble approaches.

## 4.3 EMPIRICAL ANALYSIS

**Would more "ordinary" source tasks help SESoM?** In our main experiments, we use 6 source tasks as mentioned earlier, regardless of their actual transferability given the target task. However, if a source task is too different from the target task, it is possible that this source task wouldn't effect the target task positively in general. In order to investigate how SESoM performs when less preferable source tasks are added, we conduct experiments as follows. First, given each target task, we pick top 1, top 3 and top 5 most generally helpful source tasks for this target task, following SPoT-t (Vu

Table 3: Results on target tasks with different number of few-shot training labels. All the scores are the average of 5 runs, with standard errors included in subscripts.

| Data Size | Method | WNLI | MRPC | RTE | MultiRC | BoolQ | WiC | WSC | CB | Avg |
|---|---|---|---|---|---|---|---|---|---|---|
| 64 | PPT | 48.51 | 68.53 | **69.22** | 60.54 | 62.04 | 53.59 | **62.00** | 54.29 | $59.84_{1.83}$ |
| | ATTEMPT | 52.68 | 76.76 | 55.88 | 58.87 | 62.35 | 57.74 | 48.46 | 50.36 | $57.89_{4.00}$ |
| | FW Ensemble | 53.80 | 77.87 | 65.68 | 64.78 | 72.65 | 62.66 | 50.96 | 58.21 | $63.33_{4.40}$ |
| | SESoM | **57.69** | **84.56** | 67.33 | **69.18** | **76.09** | **63.71** | 60.92 | **77.14** | $\mathbf{69.58}_{1.42}$ |
| 128 | PPT | 50.14 | 68.68 | 66.75 | 62.27 | 62.16 | 51.25 | **61.81** | 75.83 | $62.36_{2.93}$ |
| | ATTEMPT | 54.37 | 77.60 | 60.29 | 58.31 | 66.47 | 53.89 | 51.35 | 69.29 | $61.44_{5.20}$ |
| | FW Ensemble | 53.55 | 79.80 | 67.36 | 59.78 | 62.65 | 56.08 | 55.58 | 74.29 | $63.64_{4.49}$ |
| | SESoM | **58.10** | **84.61** | **68.09** | **68.87** | **75.93** | **63.73** | 61.54 | **82.86** | $\mathbf{70.46}_{1.06}$ |
| Full | PPT | 50.70 | 87.50 | **80.87** | 71.39 | 79.57 | 66.61 | 62.78 | 78.57 | $72.24_{0.01}$ |
| | ATTEMPT | 55.56 | 88.14 | 75.31 | 70.86 | 79.37 | **68.50** | 59.23 | 75.36 | $71.54_{1.89}$ |
| | FW Ensemble | 54.93 | 89.46 | 79.06 | 68.59 | 75.82 | 66.46 | 63.46 | 76.79 | $71.82_{0.84}$ |
| | SESoM | **57.75** | **89.92** | 79.42 | **72.14** | **80.79** | 67.54 | **63.50** | **86.43** | $\mathbf{74.69}_{0.25}$ |

Table 4: Results of different design choices on a pre-trained T5-Base model. All the scores are the average of 20 runs, with standard errors included in subscripts. "Ensemble acc SP" stands for Ensemble acc. source prompts. "PLG" stands for Pseudo Label Generation. Training details can be found in Appx. B.4.

| Method | WNLI | MRPC | RTE | MultiRC | BoolQ | WiC | WSC | CB | Avg |
|---|---|---|---|---|---|---|---|---|---|
| Ensemble acc SP | 52.96 | 79.69 | 64.15 | 63.92 | 74.46 | 60.97 | 52.34 | 71.25 | $64.96_{4.29}$ |
| PLG | 50.07 | 77.57 | 65.03 | 65.32 | 72.98 | 63.14 | 51.39 | 43.74 | $61.15_{6.37}$ |
| SESoM | **56.34** | **84.34** | **66.66** | **66.91** | **74.80** | **63.68** | **53.56** | **74.02** | $\mathbf{67.54}_{2.16}$ |

et al., 2022). Then we apply SESoM with the top 1, top 3 and top 5 source tasks separately, and compare with SESoM's main results with 6 default source tasks. Results are shown in Table 2. From Table 2 we can see that SESoM can achieve better performances when the number of source tasks increases, although the newly added source tasks might be less preferable to the target task in general. It indicates that for the target samples on which the generally less favourable source tasks can actually perform well, SESoM is able to set proper sample-specific weights and lean more on these source models for such specific target samples, which implies the effectiveness of SESoM sample-specific strategy.

**The effect of the number of shots.** In main experiments, the number of few-shot labeled data we use for each target task is 32 following existing work. We wonder how SESoM would perform when we increase the number of shots compared to other methods. Therefore we have tried to set the number of shots as 64, and 128 and full. Results are presented in Table 3. We can find that SESoM can consistently achieve better performance compared to existing methods when the number of few-shot labeled target data increases. It suggests that SESoM can be applied to a wide range of few-shot settings. Even in full-data settings, SESoM outperforms existing methods although not as significantly as in few-shot settings.

**Verification of other design choices.** Before the current SESoM, we have also tried some related design choices that outperform existing baselines to some extent, but turned out to be less effective than SESoM. For example, we have tried different inputs for attention module $\mathcal{G}$. Instead of using source logits as our attention module's other input given input sample $X$ in SESoM, we have tried using the source prompts as inputs, i.e., replacing $\mathbf{l}_j$ in Eq.2 with $\mathbf{P}_j$. Other components of this method stay the same as SESoM. We refer to this method as "Ensemble acc. source prompts". We have also tried directly generating pseudo labels from source models to train the target soft prompt. Specifically, we first prompt tune source prompts on the target data sets with the few-shot target training data. Then we use these prompt-tuned source prompts with the pre-trained language model, to generate pseudo labels for the entire target datasets. The final pseudo label of each target sample is decided by majority voting of the 6 source models. These pseudo labelled samples are used to train the soft prompt of the target task before few-shot prompt tuning with the few-shot target training data. We refer to this method as "Pseudo Label Generation'. Results of these methods are shown in Table 4. We can see that they achieve relatively similar performances with FW Ensemble method in Table 1, while all under-perform SESoM.

## 4.4 CASE STUDY

In this section, we conduct a case study to verify the effectiveness of SESoM's sample-specific model ensemble strategy. First, we would like to see whether it is true that given one target task, different samples of this target task require different source models. Because it is the basic assumption under which we propose SESoM's sample-specific ensemble strategy. In Table 5, we show two samples from target task MRPC. Under each sample, in the row of "preds from individual source", we show each individual source model's prediction given this sample. We can see that for Example #1, QQP and QNLI makes the correct prediction while for Example #2, MNLI, SST2 and QNLI makes the correct prediction. This sample-specific performance of different source models on target-task samples is not rare for all target tasks. More examples of other target tasks can be found in Appx. E. Moreover, the majority or at least half of source models make wrong predictions for each example. It indicates that universal model ensemble or majority vote model ensemble approaches would fail on these examples.

Second, we would like to see whether SESoM's sample-specific contribution weights of different source models generated by its attention module $\mathcal{G}$, are aligned with the performances of different source models. In Table 5 under each example, we show the contribution weights from SESoM for ensembling each source models in the row of "weights from SESoM". We can see that SESoM generally generates higher weights for source models that make the correct predictions on each example and the weight distribution is different for different examples. More examples of other target tasks can be found in Appx. E. It indicates that SESoM can generate sample-specific weights for model ensemble according to the actual sample-specific performance of different source models.

Table 5: Case study from MRPC target task. "Label" (yellow cells) is ground truth label and "Pred" (yellow cells) is the predicted label obtained by SESoM with the weights (orange cells) shown in the table. "Preds from individual source" (pink cells) shows predictions of each source model.

| Example # 1 | | | | | | | [MRPC] |
|---|---|---|---|---|---|---|---|
| s_1: Unable to find a home for him, a judge told mental health authorities they needed to find supervised housing and treatment for DeVries somewhere in California. | | | | | | | |
| s_2: The judge had told the state Department of Mental Health to find supervised housing and treatment for DeVries somewhere in California. | | | | | | | |
| **Pred** | 1 (Equivalent) | | **Source Models** | | | | | |
| **Label** | 1 (Equivalent) | | **MNLI** | **SST2** | **QNLI** | **QQP** | **SQuAD** | **ReCoRD** |
| **Preds from individual source** | | | 0 ✗ | 0 ✗ | 1 ✓ | 1 ✓ | 0 ✗ | 0 ✗ |
| **Weights from SESoM** | | | 0.0120 | 0.0708 | 0.1346 | 0.5857 | 0.1266 | 0.0702 |

| Example # 2 | | | | | | | [MRPC] |
|---|---|---|---|---|---|---|---|
| s_1: This integrates with Rational PurifyPlus and allows developers to work in supported versions of Java, Visual C # and Visual Basic.NET. | | | | | | | |
| s_2: IBM said the Rational products were also integrated with Rational PurifyPlus, which allows developers to work in Java, Visual C # and VisualBasic.Net. | | | | | | | |
| **Pred** | 1 (Equivalent) | | **Source Models** | | | | | |
| **Label** | 1 (Equivalent) | | **MNLI** | **SST2** | **QNLI** | **QQP** | **SQuAD** | **ReCoRD** |
| **Preds from individual source** | | | 1 ✓ | 1 ✓ | 1 ✓ | 0 ✗ | 0 ✗ | 0 ✗ |
| **Weights from SESoM** | | | 0.2381 | 0.2055 | 0.2486 | 0.2240 | 0.0219 | 0.0620 |

## 5 CLOSING REMARKS

In this paper we explored the potentials of prompt tuning methods in few-shot settings. We have found in our exploration that by properly transferring knowledge from trained soft prompts of source tasks, prompt tuning's few-shot performance can be significantly improved. Specifically, our proposed method, sample-specific ensemble of source models (SESoM) outperforms existing methods by a large margin in every tested few-shot scenario. SESoM adjusts the contribution of each source model for each target sample separately when ensembling source model outputs. Our empirical analysis suggests that the two key components of SESoM, the model-level ensemble instead of prompt-level fusion, and the sample-specific strategy for model ensemble, are both critical to its superior performance. SESoM also consistently achieves superior performance given larger numbers of labelled data and larger numbers of source tasks.

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

# A  DATASETS

## A.1  SOURCE TASKS

Following existing work(Asai et al., 2022), we use six tasks that have been considered generally helpful to other NLP tasks as our source tasks. They are MNLI (Williams et al., 2018), QNLI (Demszky et al., 2018), QQP (Wang et al., 2018), SST2 (Socher et al., 2013), SQuAD (Rajpurkar et al., 2016), and ReCoRD (Zhang et al., 2018). Details are shown in Table 6.

Table 6: The tasks included in source models. NLI is natural language inference, and QA is question answering.

| Dataset | \|Train\| | Category | Task | Domain | Metric |
|---------|-----------|----------|------|--------|--------|
| MNLI | 393k | GLUE | NLI | misc. | accuracy |
| SST-2 | 67k | GLUE | Sentiment analysis | Movie Reviews | accuracy |
| QQP | 364k | GLUE | Paraphrase | Social QA questions | accuracy & F1 |
| QNLI | 105k | GLUE | QA/NLI | Wikipedia | accuracy |
| SQuAD | 88k | MRQA 2019 | QA | Wikipedia | F1 & EM |
| ReCoRD | 101k | SuperGLUE | QA | News (CNN, Daily Mail) | F1 & EM |

## A.2  TARGET TASKS

We evaluate our model on the following GLUE (Wang et al., 2018) and SuperGLUE (Wang et al., 2019) tasks: WNLI (Wang et al., 2018), MRPC (Dolan et al., 2005), BoolQ (Clark et al., 2019), MultiRC (Khashabi et al., 2018), RTE (Giampiccolo et al., 2007), WiC (Pilehvar & Camacho-Collados, 2019), WSC (Levesque et al., 2012), and CB (De Marneffe et al., 2019). Details are shown in Table 7.

Table 7: The tasks included in source models. NLI is natural language inference, and QA is question answering.

| Dataset | \|Train\| | Category | Task | Domain | Metric |
|---------|-----------|----------|------|--------|--------|
| WNLI | 634 | GLUE | coreference/NLI | Fiction books | accuracy |
| MRPC | 3.7k | GLUE | Paraphrase | News | accuracy & F1 |
| RTE | 2.5k | SuperGLUE | NLI | News, Wikipedia | accuracy |
| MultiRC | 5.1k | SuperGLUE | QA | various | F1 & EM |
| BoolQ | 9.4k | SuperGLUE | QA | Google queries, Wikipedia | accuracy |
| WiC | 6k | SuperGLUE | WSD | WordNet, VerbNet, Wiktionary | accuracy |
| WSC | 554 | SuperGLUE | coreference | fiction books | accuracy |
| CB | 250 | SuperGLUE | NLI | various | accuracy & F1 |

# B IMPLEMENTATION DETAILS

## B.1 SOURCE PROMPTS TRAINING

We follow Asai et al. (2022)[2] to train the source prompts, $[\mathbf{P}_1; ...; \mathbf{P}_6]$ with corresponding source task data. These source prompts are prompt-tuned(Lester et al., 2021) respectively on MNLI (Williams et al., 2018), QNLI (Demszky et al., 2018), QQP (Wang et al., 2018), SST2 (Socher et al., 2013), SQuAD (Rajpurkar et al., 2016), and ReCoRD (Zhang et al., 2018). Details can be found in Table 6. Soft tokens size ($m$) is 100 for all pre-trained models and embedding dimensions ($d$) are 768, 1024 and 1024 respectively for T5-{base, large, 3b}. The rest hyper-parameters are shown in Table 8.

Table 8: Parameters of baseline and SESoM training. "Batch size" is different for T5-base, T5-large and T5-XL.

| Parameters | Source Prompt | SPoT_t | PPT | ATTEMPT | SESoM |
|---|---|---|---|---|---|
| max source length | 512 | 256 | 256 | 256 | 256 |
| learning rate | 3e-1 | 3e-1 | 3e-1 | 3e-1 | 3e-1 |
| train epoch | 10 | 30 | 30 | 20 | 20 |
| optimizer | AdamW | AdamW | AdamW | AdamW | AdamW |
| batch size | 32/16/4 | 32/16/4 | 32/16/4 | 32/16/8 | 32/16/4 |

## B.2 BASELINES

### B.2.1 SPoT-TARGET (SPoT-T)

We first train a randomly initialized soft prompt on a target task, which is referred as *target prompt*, using the target few-shot training data. Then we calculate the cosine similarity between the target prompt $\mathbf{P}_t$ and the source prompts, $[\mathbf{P}_1; ...; \mathbf{P}_6]$. The source prompt $\mathbf{P}_i$, which shares the highest similarity with $\mathbf{P}_t$, is used to initialize the soft prompt training. We then prompt-tune this initialized soft prompt $\hat{\mathbf{P}}_t$ on the same target few-shot training data. Finally, the source model $[\hat{\mathbf{P}}_t; \theta]$ is tested with verbalizer (Appx. B.5) on the original validation set of target task. Hyper-parameter details can be found in Table 8.

### B.2.2 PPT

For a fair comparison, we replicate Gu et al. (2022) only with pre-training a soft prompt on MNLI, QNLI, QQP, SST2, SQuAD, and ReCoRD which we used for training source prompts. Following Gu et al. (2022), we firstly unify six source tasks to a single format——multiple-choice classification. For example, CB is formatted as,

```
#premise.    #hypothesis. A. yes. B. maybe. C. no. The correct one is
```

Classification text labels of CB is `A(no)`, `B(maybe)` and `C(yes)`. We then prompt-tune a randomly initialized soft prompt $\mathbf{P}_{ppt}$ on these unified datasets. We use the same hyperparametes as training SPoT-t. Then for each target, $\mathbf{P}_{ppt}$ is tuned on the same $D_{train}$ and $D_{dev}$ to get target-specific soft prompt $\hat{\mathbf{P}}_{ppt}$. Finally, the source model $[\hat{\mathbf{P}}_{ppt}; \theta]$ is tested with verbalizer (Appx. B.5) applied on the same $D_{test}$. See training details in Table 8.

### B.2.3 ATTEMPT

Following Asai et al. (2022), $[\mathbf{P}_1; ...; \mathbf{P}_6]$ (Appx. B.1) are used as source prompts. ATTEMPT first initialize a attention module between source prompts and inputs. ATTEMPT interpolates the source prompts and a newly initialized target-task-specific prompt $\mathbf{P}_{target}$ given attention scores generated by the attention module to produce a target soft prompt $\mathbf{P}_t$, then $[\mathbf{P}_t; \theta]$ is used to produce predictions. We use the same training parameters used in Asai et al. (2022), which is shown in Table 8.

---

[2]https://github.com/AkariAsai/ATTEMPT

## B.3 SESoM

The hyper-parameters of $\mathcal{G}$ are shown in Table 9.

Table 9: Dimensions of $\mathcal{G}$ of SESoM used in Table 1.

| Model | Method | WNLI | MRPC | RTE | MultiRC | BoolQ | WiC | WSC | CB |
|---|---|---|---|---|---|---|---|---|---|
| T5-base | $d'_x$ | 32 | 32 | 100 | 100 | 64 | 100 | 64 | 128 |
| | $d'_l$ | 32 | 32 | 100 | 100 | 64 | 100 | 100 | 128 |
| | $d'$ | 32 | 128 | 100 | 100 | 64 | 100 | 100 | 128 |
| | $\mathcal{G}$'s dropout % | 50 | 0 | 0 | 0 | 0 | 0 | 0 | 0 |
| T5-large | $d'_x$ | 100 | 64 | 64 | 32 | 100 | 100 | 32 | 64 |
| | $d'_l$ | 100 | 64 | 256 | 32 | 64 | 64 | 32 | 64 |
| | $d'$ | 100 | 64 | 512 | 128 | 100 | 100 | 64 | 256 |
| | $\mathcal{G}$'s dropout % | 50 | 0 | 0 | 50 | 0 | 0 | 50 | 0 |
| T5-XL | $d'_x$ | 100 | 32 | 32 | 100 | 64 | 64 | 64 | 64 |
| | $d'_l$ | 100 | 64 | 64 | 100 | 128 | 64 | 64 | 64 |
| | $d'$ | 100 | 64 | 64 | 100 | 256 | 512 | 64 | 512 |
| | $\mathcal{G}$'s dropout % | 50 | 0 | 50 | 0 | 0 | 0 | 50 | 0 |

## B.4 Ablation Study

Before SESoM, we also tried some related methods that turned out to be less effective.

**Ensemble acc SP.** Firstly, we tried consider to use input-prompt attention, where max-pool embedding input $\hat{\mathbf{x}}$ is used as key and source prompts $[\mathbf{P}_1; ...; \mathbf{P}_6]$ are values to calculate attention score to computing a linear combination of pre-softmax logits. The architecture design of the attention module is the same with SESoM. The dimensions of the attention module after hyper-parameters tuning is shown in Table 10.

Table 10: Hyper-parameters of $\mathcal{G}$ of "Ensemble acc SP" used in Table 4

| Model | Method | WNLI | MRPC | RTE | MultiRC | BoolQ | WiC | WSC | CB |
|---|---|---|---|---|---|---|---|---|---|
| T5-base | $d'_x$ | 32 | 32 | 128 | 128 | 64 | 100 | 64 | 100 |
| | $d'_l$ | 32 | 32 | 128 | 128 | 64 | 100 | 64 | 100 |
| | $d'$ | 32 | 64 | 128 | 128 | 128 | 100 | 64 | 100 |
| | $\mathcal{G}$'s dropout % | 50 | 0 | 0 | 0 | 0 | 0 | 0 | 0 |

**Majority vote.** Secondly, we consider whether predictions from each source model $[\mathbf{P}_1; \theta], ..., [\mathbf{P}_6; \theta]$ can provide enough information for final prediction. We get the pre-softmax logits $\mathbf{L}_x = [\mathbf{l}_{x,1}; ...; \mathbf{l}_{x,6}]$ then conduct verbalizer mapping to obtain $\hat{\mathbf{L}}_x = [\hat{\mathbf{l}}_{x,1}; ...; \hat{\mathbf{l}}_{x,6}]$. Then for each input, we obtain 6 predicted labels $l_1, ..., l_6$ from $\hat{\mathbf{L}}_x$. The majority vote of $l_1, ..., l_6$ is used as final prediction. For tie condition, we uniformly sample one of them as prediction.

## B.5 Verbalizer Details

Source prompts are trained on six source datasets in a text-to-text format. Different source datasets have different verbalizers. For example, CB (De Marneffe et al., 2019) is classified as "entailment", "neutral" and "contradiction"; and MRPC (Dolan et al., 2005) is classified as "equivalent" and "not equivalent". We show the verbalizers used for all target task in Table 11. For rare cases in which the source model after few-shot prompt tuning is not able to transfer all the verbalizers from source dataset to target dataset, for all baselines and our method, we conduct a *verbalizer mapping* for pre-softmax logits.

Based on original verbalizers in Table 11, we develop a mapping dictionary $\mathbf{M_T}$ to map text to the token for target dataset $T$. For example, we map "True" to "1", and "False" to "0" for the prediction of source models with source prompts trained on Squad(Rajpurkar et al., 2016) and Record(Zhang et al., 2018). Formally, for the $i$th source model, the pre-softmax logit $\mathbf{l}_i = [\mathbf{l}_{i,1}; ...; \mathbf{l}_{i,v}]$ obtained

from source model $[\mathbf{P}_i; \theta]$ on target dataset $T$ is transformed to $\hat{\mathbf{l}}_i$ by Algorithm 1, where $v$ is the vocabulary size of tokens. More specifically, all the logits in each pre-softmax logit of $\mathbf{l}_i$ will be swapped to the same position. For example, in source model of ReCoRD, the pre-softmax logit of token 209 in $\hat{\mathbf{l}}_i$ is replaced with the maximum of the pre-softmax logit of token 209 and 10998 in $\mathbf{l}_i$, because $\mathbf{M_T}[10998] = 209$, where 10998 is the first token representing "True" and 209 is the first token representing "1".

Table 11: Mapping dictionary details.

| Target | Label | |
|---|---|---|
| | # | Text |
| WNLI | 0 | not_entailment |
| | 1 | entailment |
| MRPC | 0 | not_equivalent |
| | 1 | equivalent |
| RTE | 0 | entailment |
| | 1 | not_entailment |
| BoolQ | 0 | False |
| | 1 | True |
| MultiRC | 0 | False |
| | 1 | True |
| WiC | 0 | False |
| | 1 | True |
| WSC | 0 | False |
| | 1 | True |
| CB | 0 | entailment |
| | 1 | contradiction |
| | 2 | neutral |

---

**Algorithm 1:** Mapping Algorithm

---

**Data:** $\mathbf{M_T}, \mathbf{l}_i$
**Result:** $\hat{\mathbf{l}}_i$
$\hat{\mathbf{l}}_i \leftarrow \mathbf{l}_i$;
**for** $k \in \mathbf{M_T}$ **do**
    $\hat{\mathbf{l}}_{i,k} \leftarrow \min(\mathbf{l}_{i,k}, \mathbf{l}_{i,\mathbf{M_T}[k]})$ ;
    $\hat{\mathbf{l}}_{i,\mathbf{M_T}[k]} \leftarrow \max(\mathbf{l}_{i,k}, \mathbf{l}_{i,\mathbf{M}[k]})$;
**end**

---

### B.6 INFERENCE TIME

Compared to non-ensemble baselines on the same pre-trained language model, SESoM indeed increases inference time due to the multiple forward passes of source models. In order to achieve a fair comparison, we compare SESoM to prompt fusion model (ATTEMPT(Asai et al., 2022)), which uses the same number of source tasks as SESoM. As a sample-specific knowledge fusion method, ATTEMPT fuses $N$ soft prompts of source tasks together to construct a new prompt for each target sample and foward it together with the target sample to the pre-trained language model, which is one forward pass per sample for inference. While SESoM runs $N$ forward passes for each sample to obtain the source logits.

Although SESoM requires more forward passes given one target sample, it requires a smaller pre-trained language model (shorter inference time per forward pass) to achieve the same /or even better performance. SESoM with a small pre-trained model (shorter inference time per forward pass) can outperform non-ensemble baseline with a much larger pre-trained model (longer inference time per forward pass). For example, SESoM with T5-base outperforms ATTEMPT with T5-XL on few-shot classification (please see Table 1). The inference time of SESoM with T5-base is 31.38s while the inference of ATTEMPT with T5-XL takes 38.13s, which shows that the inference time of SESoM with T5-base is shorter. The inference time is the average of 20 runs with different random seeds on

all $8$ target tasks. It indicates that SESoM can achieve a better performance with shorter inference time.

## C   ANALYSIS ON ATTENTION

Figure 2 shows the average attention weights of $\mathcal{G}$ between source and target tasks on $D_{\text{test}}$. The average weights are produced by the SESoM model with T5-base and hyper-parameters are shown in Appx. B.3. $\mathcal{G}$ has higher attention score between more similar target and source tasks. $\mathcal{G}$ gives significantly higher attentions to QQP for MRPC and WSC or MNLI for RTE and WNLI. MRPC and QQP are both tasks to determine whether a pair of questions/sentences are semantically equivalent. WSC is coreference resolution task to determine the correct referent of the pronoun from the provided noun, which shares similarities with QQP as well. Besides, WNLI, RTE,and MNLI are all textual entailment challenges, which share similarities between each other.

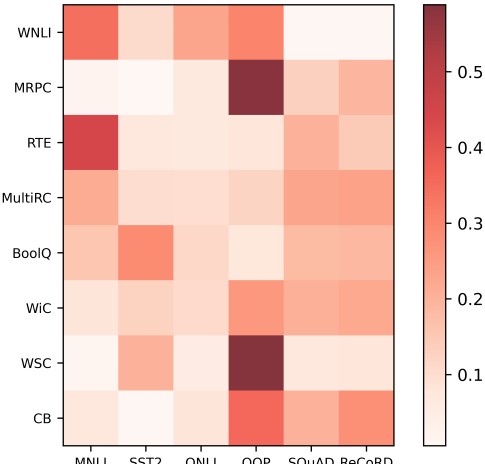

Figure 2: Average attention weights of $\mathcal{G}$.

We also find the attention weights and F1 scores are highly positive correlated on MRPC, RTE, MultiRC, WSC and CB tasks. Pearson product-moment correlation coefficients between attention weights and F1 scores are shown as follows,

| Target Task | WNLI | MRPC | RTE | MultiRC | BoolQ | WiC | WSC | CB |
|---|---|---|---|---|---|---|---|---|
| Corr. coeff. | -0.29 | 0.42 | 0.44 | 0.46 | 0.03 | 0.75 | 0.63 | 0.45 |

High F1 scores of source model on target task also indicate underlying task similarities. Fixed-Weight ensemble model takes advantage of these F1 scores to get a weighted average of all source pre-max logits to make predictions. But as we showed in Table 1 and Section 4.2, these weights are not enough to obatin the best performance on target tasks.

# D  MORE RESULTS

## D.1  EFFECT OF THE NUMBER OF SHOTS

Besides our main experiments (Table 3), we also reported results on target tasks with 8 training samples in Table 12. The performance of SESoM is better than all the baselines with 32 training samples, which are shown in Table 1.

Table 12: Results on target tasks with 8 samples of few-shot training labels. All the scores are the average of 10 runs, with standard errors included in subscripts.

| Data Size | Method | WNLI | MRPC | RTE | MultiRC | BoolQ | WiC | WSC | CB | Avg |
|---|---|---|---|---|---|---|---|---|---|---|
| 8 | FW Ensemble | 53.73 | 71.10 | 58.70 | 63.17 | 72.44 | 63.17 | 51.44 | 61.52 | $61.90_{3.97}$ |
|  | SESoM | 58.31 | 78.01 | 63.65 | 66.91 | 71.27 | 62.73 | 54.97 | 66.61 | $65.31_{3.14}$ |

## D.2  EFFECT OF PROMPT-TUNING SOURCE MODELS

In main experiments, we train each source model with the few-shot target data in a prompt-tuning manner. We wonder how SESoM would perform when we skip this step. Therefore, we tried to perform SESoM on non-tuned source models. Results are presented in Table 13. We can find that with only 8 and 32 samples of few-shot training samples, SESoM with prompt-tuned source models cannot achieve significantly better performance than SESoM with non-tuned source models. However, when the number of few-shot labeled target data increase, SESoM with prompt-tuned source models achieve better performance, because the source models themselves have a better performance on target datasets.

Table 13: Results on target tasks with different numbers of few-shot training samples. "SESoM w/ PT" represents SESoM with prompt-tuned source models; "SESoM w/o PT" is SESoM with non-tuned source models. All the scores of data size of 8, 32 and 128 are the average of 10, 10 and 5 runs, with standard errors included in subscripts.

| Data Size | Method | WNLI | MRPC | RTE | MultiRC | BoolQ | WiC | WSC | CB | Avg |
|---|---|---|---|---|---|---|---|---|---|---|
| 8 | SESoM w/o PT | 58.24 | 76.34 | 63.16 | 66.45 | 69.50 | 63.08 | 54.97 | 67.41 | $64.89_{3.46}$ |
|  | SESoM w. PT | 58.31 | 78.01 | 63.65 | 66.91 | 71.27 | 62.73 | 54.97 | 66.61 | $65.31_{3.14}$ |
| 32 | SESoM w/o PT | 55.21 | 82.57 | 65.20 | 63.53 | 74.89 | 62.74 | 53.37 | 75.18 | $66.59_{3.81}$ |
|  | SESoM w. PT | 58.31 | 84.22 | 66.86 | 66.72 | 72.87 | 64.20 | 53.56 | 72.86 | $67.45_{2.41}$ |
| 128 | SESoM w/o PT | 58.24 | 83.61 | 65.89 | 63.89 | 75.74 | 63.52 | 55.29 | 74.29 | $67.57_{2.57}$ |
|  | SESoM w. PT | 58.10 | 84.61 | 68.09 | 68.87 | 75.93 | 63.73 | 61.54 | 82.86 | $70.46_{1.06}$ |

## D.3  EFFECT OF ENSEMBLE

We wonder whether our ensemble method is better than choosing single most appropriate source model's prediction as final prediction. We implement a *hard variant*, where after attention weights are computed, we set the argmax position to 1 and all remaining entries to 0s. Results are presented in Table 14. It shows that our ensemble attention is useful compared to selecting the single most appropriate source model.

Table 14: Results on target tasks with 32 samples of few-shot training labels with different ensemble. All the scores are the average of 20 runs, with standard errors included in subscripts.

| Method | WNLI | MRPC | RTE | MultiRC | BoolQ | WiC | WSC | CB | Avg |
|---|---|---|---|---|---|---|---|---|---|
| Hard Variant | 48.52 | 73.00 | 54.49 | 53.58 | 60.53 | 56.17 | 49.18 | 56.52 | $56.50_{7.66}$ |
| SESoM | 56.34 | 84.34 | 66.66 | 66.91 | 74.80 | 63.68 | 53.56 | 74.02 | $67.54_{2.16}$ |

# E CASE STUDY

Table 15: Case study from WSC and MultiRC target task. "Label" (yellow cells) is ground truth label and "Pred" (yellow cells) is the predicted label obtained by SESoM with the weights (orange cells) shown in the table. "Preds from individual source" (pink cells) shows predictions of each source model $[\mathbf{P}_{1,t};\theta], ..., [\mathbf{P}_{6,t};\theta]$ obtained by the pre-softmax logits $[\mathbf{l}_{x,1}; ...; \mathbf{l}_{x,T}]$.

| Example # 1 | | | | | | | [WSC] |
|---|---|---|---|---|---|---|---|
| text: **Alice** tried frantically to stop her daughter from barking at the party, leaving us to wonder why **she** was behaving so strangely. | | | | | | | |
| s_1: Alice | | | | | | | |
| s_2: She | | | | | | | |

| Pred | 0 (False) | Source Models | | | | | |
|---|---|---|---|---|---|---|---|
| Label | 0 (False) | MNLI | SST2 | QNLI | QQP | SQuAD | ReCoRD |
| Preds from individual source | | 1 ✗ | 0 ✓ | 1 ✗ | 0 ✓ | 1 ✗ | 1 ✗ |
| Weights from SESoM | | 0.0480 | 0.3348 | 0.0912 | 0.3270 | 0.0984 | 0.1005 |

| Example # 2 | | | | | | | [WSC] |
|---|---|---|---|---|---|---|---|
| text: Well satisfied with his purchases and feeling very elegant indeed, Babar goes to the **photographer** to have **his** picture taken. | | | | | | | |
| s_1: photographer | | | | | | | |
| s_2: his | | | | | | | |

| Pred | 0 (False) | Source Models | | | | | |
|---|---|---|---|---|---|---|---|
| Label | 0 (False) | MNLI | SST2 | QNLI | QQP | SQuAD | ReCoRD |
| Preds from individual source | | 1 ✗ | 1 ✗ | 1 ✗ | 0 ✓ | 1 ✗ | 1 ✗ |
| Weights from SESoM | | 0.0732 | 0.0285 | 0.0972 | 0.5973 | 0.1048 | 0.0991 |

| Example # 3 | | | | | | | [MultiRC] |
|---|---|---|---|---|---|---|---|
| question: Sarah introduces him to three other guests. Name them. | | | | | | | |
| paragraph: The bar was manned by an expensive humanoid robot. It turned toward Sarah's wave and acknowledged her with a nod, moments later setting a fluted glass of sparkling liquid in front of her. I marveled at the robot's smoothness and coordination. Clearly, it was a high-end model. Sarah transferred the glass to my free hand and pulled me away from the bar for more introductions, with Alexis trailing after us. I spent the evening listening, mostly. Listening and stuffing my face with all the bits of fine food provided. No one minded; Sarah's inner circle was content to fill our circle of couches with plenty of chatter. Ray, a plump man who was grey where he wasn't bald. Zheng, short and dark and lean, with a very intense gaze. He made me a little uncomfortable. Kishori, petite, her hair strung out in a series of braids that reached nearly to her waist. | | | | | | | |
| answer: Ray, Zheng, and Kishori | | | | | | | |

| Pred | 1 (True) | Source Models | | | | | |
|---|---|---|---|---|---|---|---|
| Label | 1 (True) | MNLI | SST2 | QNLI | QQP | SQuAD | ReCoRD |
| Preds from individual source | | 1 ✓ | 0 ✗ | 0 ✗ | 0 ✗ | 1 ✓ | 1 ✓ |
| Weights from SESoM | | 0.2094 | 0.1028 | 0.1098 | 0.1019 | 0.2377 | 0.2383 |

| Example # 4 | | | | | | | [MultiRC] |
|---|---|---|---|---|---|---|---|
| question: What were Zheng's traits? | | | | | | | |
| paragraph: Same with **Example # 3**. | | | | | | | |
| answer: Humanoid | | | | | | | |

| Pred | 0 (False) | Source Models | | | | | |
|---|---|---|---|---|---|---|---|
| Label | 0 (False) | MNLI | SST2 | QNLI | QQP | SQuAD | ReCoRD |
| Preds from individual source | | 0 ✓ | 0 ✓ | 1 ✗ | 1 ✗ | 1 ✗ | 1 ✗ |
| Weights from SESoM | | 0.2347 | 0.1932 | 0.1891 | 0.1145 | 0.1386 | 0.1300 |

We show more examples of attention weights on samples from different target task in Table 15, Table 16 and Table 17. They show the superior power of SESoM to give higher attention to those pre-softmax logits, which prefers correct label.

Table 16: Case study from BoolQ and WiC target task. "Label" (yellow cells) is ground truth label and "Pred" (yellow cells) is the predicted label obtained by SESoM with the weights (orange cells) shown in the table. "Preds from individual source" (pink cells) shows predictions of each source model $[\mathbf{P}_{1,t};\theta], ..., [\mathbf{P}_{6,t};\theta]$ obtained by the pre-softmax logits $[\mathbf{l}_{x,1}; ...; \mathbf{l}_{x,T}]$.

| Example # 5 | | | | | | [BoolQ] |
|---|---|---|---|---|---|---|
| question: Did the stock market crash of 1929 caused the great depression? | | | | | | |
| passage: Wall Street Crash of 1929 – The Wall Street Crash of 1929, also known as Black Tuesday, the Great Crash, or the Stock Market Crash of 1929, began on October 24, 1929, and was the most devastating stock market crash in the history of the United States, when taking into consideration the full extent and duration of its after effects. The crash, which followed the London Stock Exchange's crash of September, signalled the beginning of the 12-year Great Depression that affected all Western industrialized countries. | | | | | | |

| **Pred** | 1 (True) | **Source Models** | | | | | |
|---|---|---|---|---|---|---|---|
| **Label** | 1 (True) | **MNLI** | **SST2** | **QNLI** | **QQP** | **SQuAD** | **ReCoRD** |
| **Preds from individual source** | | 1 ✓ | 0 ✗ | 1 ✓ | 0 ✗ | 1 ✓ | 1 ✓ |
| **Weights from SESoM** | | 0.0194 | 0.0036 | 0.0073 | 0.0627 | 0.4285 | 0.4785 |

| Example # 6 | | | | | | [BoolQ] |
|---|---|---|---|---|---|---|
| question: Do house and cuddy get back together in season 8? | | | | | | |
| passage: Lisa Cuddy – The relationship between House and Cuddy is known by the portmanteau term, Huddy. Cuddy has what USA Today's Peter Johnson terms a cat-and-mouse relationship with House. Edelstein has described it as "a really complicated, adult relationship", explaining: "These are people who have very full lives and lots of responsibilities that perhaps conflict with their feelings for each other." The actress "would love for them to have a (romantic) relationship, because it could be as complicated as the rest of their relationship", however, she is unsure how it would affect the dynamics of the show. Jacobs commented at the end of the show's third season: "I can't see them pairing them in a permanent fashion. But they are close; they have gone through a lot together. Might there be a moment of weakness in which the two might explore their chemistry? Maybe." Questioned at the end of the fourth season on whether Cuddy and House. | | | | | | |

| **Pred** | 0 (False) | **Source Models** | | | | | |
|---|---|---|---|---|---|---|---|
| **Label** | 0 (False) | **MNLI** | **SST2** | **QNLI** | **QQP** | **SQuAD** | **ReCoRD** |
| **Preds from individual source** | | 1 ✗ | 0 ✓ | 1 ✗ | 0 ✓ | 0 ✓ | 0 ✓ |
| **Weights from SESoM** | | 0.0100 | 0.3151 | 0.0176 | 0.0878 | 0.3977 | 0.1717 |

| Example # 7 | | | | | | [WiC] |
|---|---|---|---|---|---|---|
| s_1: An emerging professional **class**. | | | | | | |
| s_2: Apologizing for losing your temper, even though you were badly provoked, showed real **class**. | | | | | | |

| **Pred** | 0 (False) | **Source Models** | | | | | |
|---|---|---|---|---|---|---|---|
| **Label** | 0 (False) | **MNLI** | **SST2** | **QNLI** | **QQP** | **SQuAD** | **ReCoRD** |
| **Preds from individual source** | | 1 ✗ | 1 ✗ | 1 ✗ | 0 ✓ | 0 ✓ | 0 ✓ |
| **Weights from SESoM** | | 1.6376e−4 | 2.4063e−4 | 2.1660e−4 | 0.2839 | 0.3430 | 0.3723 |

| Example # 8 | | | | | | [WiC] |
|---|---|---|---|---|---|---|
| s_1: He could not **touch** the meaning of the poem. | | | | | | |
| s_2: Helen Keller felt the physical world by **touch**ing people and objects around her. | | | | | | |

| **Pred** | 0 (False) | **Source Models** | | | | | |
|---|---|---|---|---|---|---|---|
| **Label** | 0 (False) | **MNLI** | **SST2** | **QNLI** | **QQP** | **SQuAD** | **ReCoRD** |
| **Preds from individual source** | | 1 ✗ | 0 ✓ | 1 ✗ | 0 ✓ | 0 ✓ | 0 ✓ |
| **Weights from SESoM** | | 0.0375 | 0.0904 | 0.0121 | 0.1120 | 0.4110 | 0.3371 |

Table 17: Case study from CB and RTE target task. "Label" (yellow cells) is ground truth label and "Pred" (yellow cells) is the predicted label obtained by SESoM with the weights (orange cells) shown in the table. "Preds from individual source" (pink cells) shows predictions of each source model $[\mathbf{P}_{1,t}; \theta], ..., [\mathbf{P}_{6,t}; \theta]$ obtained by the pre-softmax logits $[\mathbf{l}_{x,1}; ...; \mathbf{l}_{x,T}]$.

| Example # 9 | | | | | | | [CB] |
|---|---|---|---|---|---|---|---|
| premise: Who knows? The point is, do we go with it or not? Do we assume there is a shipment? hypothesis: there is a shipment. | | | | | | | |
| **Pred** | 2 (Neutral) | | | **Source Models** | | | |
| **Label** | 2 (Neutral) | **MNLI** | **SST2** | **QNLI** | **QQP** | **SQuAD** | **ReCoRD** |
| **Preds from individual source** | | 0 ✗ | 0 ✗ | 1 ✗ | 0 ✗ | 2 ✓ | 2 ✓ |
| **Weights from SESoM** | | 0.0310 | 0.0260 | 1.8867e−4 | 9.433e−5 | 0.4084 | 0.5342 |

| Example # 10 | | | | | | | [CB] |
|---|---|---|---|---|---|---|---|
| premise: But what we may not know is just what makes somebody a sucker. What makes people blurt out their credit-card numbers to a caller they've never heard of? Do they really believe that the number is just for verification and is simply a formality on the road to being a grand-prize winner? hypothesis: The number is just for verification and is simply a formality on the road to being a grand-prize winner. | | | | | | | |
| **Pred** | 1 (Contradiction) | | | **Source Models** | | | |
| **Label** | 1 (Contradiction) | **MNLI** | **SST2** | **QNLI** | **QQP** | **SQuAD** | **ReCoRD** |
| **Preds from individual source** | | 1 ✓ | 0 ✗ | 1 ✓ | 0 ✗ | 1 ✓ | 1 ✓ |
| **Weights from SESoM** | | 0.7075 | 2.1385e−3 | 0.1998 | 1.2705e−4 | 0.0234 | 0.0668 |

| Example # 11 | | | | | | | [RTE] |
|---|---|---|---|---|---|---|---|
| premise: Dana Reeve, the widow of the actor Christopher Reeve, has died of lung cancer at age 44, according to the Christopher Reeve Foundation. hypothesis: Christopher Reeve had an accident. | | | | | | | |
| **Pred** | 1 (not_entailment) | | | **Source Models** | | | |
| **Label** | 1 (not_entailment) | **MNLI** | **SST2** | **QNLI** | **QQP** | **SQuAD** | **ReCoRD** |
| **Preds from individual source** | | 1 ✓ | 0 ✗ | 1 ✓ | 0 ✗ | 1 ✓ | 1 ✓ |
| **Weights from SESoM** | | 0.5337 | 0.0092 | 0.2753 | 0.0560 | 0.0825 | 0.0434 |

| Example # 12 | | | | | | | [RTE] |
|---|---|---|---|---|---|---|---|
| premise: Hands Across the Divide was formed in March 2001, and one of its immediate aims was to press for more freedom of contact and communication right away between the two parts of Cyprus, and for early progress towards a solution to "the Cyprus problem". hypothesis: Cyprus was divided into two parts in March 2001. | | | | | | | |
| **Pred** | 1 (not_entailment) | | | **Source Models** | | | |
| **Label** | 1 (not_entailment) | **MNLI** | **SST2** | **QNLI** | **QQP** | **SQuAD** | **ReCoRD** |
| **Preds from individual source** | | 0 ✗ | 0 ✗ | 1 ✓ | 0 ✗ | 0 ✗ | 1 ✓ |
| **Weights from SESoM** | | 0.0698 | 0.1879 | 0.3535 | 0.1364 | 0.0259 | 0.2265 |

