# OpenReview forum: "Model ensemble instead of prompt fusion: a sample-specific knowledge transfer method for few-shot prompt tuning"
_ICLR.cc/2023/Conference — ICLR 2023 poster_

### Official Review · Reviewer_bhqg · 2022-10-22

**Confidence:** 4
**Correctness:** 4
**Technical Novelty And Significance:** 3
**Empirical Novelty And Significance:** 3
**Recommendation:** 8

**Clarity, Quality, Novelty And Reproducibility:**

Well written paper, with problem stated and motivations explained.  This work is orthogonal to the work like chain-of-thoughts, which improve prompt generation itself.  Instead, this method improves data efficiency with both an ensemble and improved transfer learning.  I believe it can be reproduced.

**Strength And Weaknesses:**

Strengths:
1. A method developed based on well though through motivations.
2. Empirical study shows better results.

Weaknesses:
1. To ensure this method work, the source models are expected to be robust.  In other words, some robust source models must be trained to be borrow.  But it may not be true that these source models are always available and comparable.
2. Can the source models trained on top of other pre-trained models?


**Summary Of The Paper:**

This work aims to improve the effectiveness of the prompting method.  The prompt tuning approaches have attracted growing interest due to its parameter efficiency. However the limited training samples in few-shot settings, prompt tuning fails to match the performance of full-model fine-tuning.  The authors design an ensemble method to deal with few shots and improve the effectiveness of transfer learning.  The empirical study shows promising results.


**Summary Of The Review:**

Prompt tuning is a hot topic since 2021. There have been several “hacks” showing promising results,  But realistically, there were so many unreported failure cases.  It is good to see a paper using principled approaches to systematically improve performance.  Since I have worked on this subject with several interns, I know the challenges and appreciate the improvements that this work could achieve.

My question is how the source models are obtained, and given a novel task, how can you identify the most useful source models from which to transfer information.  If source models are not available, what is your fall back scheme?

---

> ### Author Response · Authors · 2022-11-16
> **Response to Reviewer bhqg**
>
> We thank the reviewer for their time and efforts. We are encouraged by their analysis of this work's significance.
>  We try to address the reviewer’s questions below.
>
> &nbsp;&nbsp;&nbsp;&nbsp;&nbsp;&nbsp;&nbsp;&nbsp;&nbsp;_1. In order for SESoM to work, some robust source models must exist so that SESoM can take use of them. But it might not be true that these source models are always available and comparable._
>
> Our reply: In order for SESoM to work, we don’t assume to have access to pre-trained source prompts or source models.  Given a pre-trained language model, and multiple source tasks with abundant training data,  we first train the source models with the source training data, as described in Fig. 1(a). Then after obtaining such source models trained on the pre-trained language model we picked, we conduct knowledge transfer with SESoM on the same pre-trained language model given few-shot target samples.
>
> &nbsp;&nbsp;&nbsp;&nbsp;&nbsp;&nbsp;&nbsp;&nbsp;&nbsp;_2. Can the source models be trained on top of other pre-trained models?_
>
> If we understand correctly, the reviewer’s question is whether the source models trained on top of other different pre-trained language models, can be used in SESoM to train a target model based on the current pre-trained language model.  We appreciate this question raised by the reviewer. The transferability of hard/soft prompts from one pre-trained language model to other pre-trained models, is another quite active field regarding the efficient applications of huge pre-trained models. Although there are some existing work [1,2,3,4] that assess such transferability, it seems that so far there is no preliminary consensus in the community on what is the most promising direction to solve this problem.   The transferability of hard/soft prompts across different pre-trained models is exactly the future direction in our mind. In this paper, SESoM doesn’t have to face this issue because we train our source models on top of the same pre-trained model we use for target tasks.
>
> &nbsp;&nbsp;&nbsp;&nbsp;&nbsp;&nbsp;&nbsp;&nbsp;&nbsp;_3. Given a novel task, how can you identify the most useful source models from which to transfer information?_
>
>
> We have found empirically that SESoM is also able to capture the general importance of different source tasks/models correctly. The empirical analysis is shown in _Appendix C_. If we average the target-sample-specific source-model weights of all target samples, we can find that SESoM gives higher average scores on source tasks that are more similar to the target task. Therefore, if we were given too many source tasks/models and we just want to select a few to reduce the method’s inference time, we can rank and pick the source models according to these average attention scores.
>
> Moreover,  if we don’t have efficiency concerns and don’t need to reduce inference time, we have shown in Section 4.3 (first paragraph) that more source models can actually make SESoM perform better, even when the newly-added source models are not considered the optimal knowledge sources of the target task.
>
> References:
>
>
> [1] Su, Yusheng, et al. "On transferability of prompt tuning for natural language processing." Proceedings of the 2022 Conference of the North American Chapter of the Association for Computational Linguistics: Human Language Technologies. 2022.
>
> [2] Xu, Lei, et al. "Exploring the Universal Vulnerability of Prompt-based Learning Paradigm." arXiv preprint arXiv:2204.05239 (2022).
>
> [3] Shin, Taylor, et al. "AutoPrompt: Eliciting Knowledge from Language Models with Automatically Generated Prompts." Proceedings of the 2020 Conference on Empirical Methods in Natural Language Processing (EMNLP). 2020.
>
> [4] He, Junxian, et al. "Towards a Unified View of Parameter-Efficient Transfer Learning." International Conference on Learning Representations. 2021.

---

### Official Review · Reviewer_eQBR · 2022-10-24

**Confidence:** 4
**Correctness:** 4
**Technical Novelty And Significance:** 3
**Empirical Novelty And Significance:** 3
**Recommendation:** 6

**Clarity, Quality, Novelty And Reproducibility:**

Clarity
=====
The paper is clear for the most part. Some minor comments on this front below:

As mentioned already, please provide details for each dataset used both for the source and target tasks, e.g. explain the acronyms in the task names, what is the task in each case, how much training data is available for each source task, and so on. Why are the particular tasks from GLUE and Super GLUE chosen as target tasks? Why not use all of them instead of picking a subset?

In Section 3.2.1, 2 lines above Equation 2, I wasn’t sure what the notation meant of the horizontal bar between X and h_x.

In Algorithm 1 (In the Appendix), M should be M_T? Write a description of this algorithm that explains in a few words what is happening (e.g. in the min and max operations), for better readability.


Novelty
======
The specific approach for ensembling prompt-tuned source models is novel to the best of my knowledge.

Reproducibility
===========
Hyperparameter details are reported in the Appendix that can help with this. I encourage the authors to release their code as well.

Quality detailed comments
=====================
In Table 3, why are some of the previous methods and baselines missing? Also, when it comes to varying shots, it would be also interesting to report results on fewer than 32 shots. In vision benchmarks, even 1-shot and 5-shot settings are commonly considered.

Regarding efficiency, aside from the parameter efficiency discussed in the paper, computational efficiency at inference time is also a potential concern. Ensembling requires forward-passing through the language model several times (once for each prefix). It would be useful to also comment on this aspect, and the relationship to previous knowledge transfer works in terms of this.

It would be useful to also report results for another ablation: applying the proposed model without first finetuning the prefix of each source model on the target data (that is, directly using the prefix that was trained for each source task, and applying the attention model on logits produced with those prefixes).

Another interesting analysis would be comparing to a ‘hard ensembling’ variant: after attention weights are computed, set the argmax position to 1 and all remaining entries to 0s. If this hard variant performs worse than the proposed method, it would show that ensembling is useful compared to selecting the single most appropriate source model.

Minor comments
==============
‘Parameter efficiency’ doesn’t really belong in the ‘Empirical Analysis’ section since there’s nothing empirically determined in that discussion. Maybe this discussion is better suited in the Methods section.

In Figure 1, it would be visually nicer to align the diagrams showing for the different source tasks. Or is there some reason I’m missing for why some are more indented than others?


**Strength And Weaknesses:**

Strengths
========
[+] The paper is clearly written and easy to understand

[+] The proposed method outperforms the previous approaches and baselines considered on a variety of datasets

[+] The authors empirically verify different design choices and baselines and perform interesting analyses (like the effect of using the top-k most similar source domains for each target task)

Weaknesses
===========
[-] Some useful ablations / exploration of alternative design choices still missing (see below)

[-] The proposed method requires several forward passes through the language model (one per source task, that uses a different prefix) in order to perform a single prediction, thus adding computational overhead at test time compared to other methods. Would be good to add some discussion on this.

[-] While the paper is clearly written for the most part, I would have liked to see a detailed description of the tasks used, both as source tasks and the target tasks considered. I couldn’t find details of them in the Appendix either, though please let me know if I missed it. This is important as understanding the tasks considered is useful for interpreting the results.


**Summary Of The Paper:**

This paper proposes a new approach for transferring knowledge from source tasks to a target task that has only a few examples available. Specifically, they use source models based on a pre-trained (and frozen) language model, where each source model consists of a trained prefix (obtained via prefix tuning on that source task). Their approach is the following: first, they finetune the prefix of each source task using data from the target task. Then, keeping all parameters frozen (both the language model as well as the per-source learned prefixes), they train an attention model which, given (the tokens of) a (max-pooled) input sequence representing a single example from the target domain, and given the logits produced by each source model for the target task, figures out how to ensemble those logits on a per-example basis, so as to do best on that target example. After having trained the attention model on the few-shot data from the target task, it is deployed to make predictions for other examples from the target task. They show empirically that their method outperforms previous approaches for transferring knowledge that weren’t explicitly designed for the few-shot setting. They also demonstrate that they outperform ensembling baselines, that their approach can work well for larger numbers of shots too, and they verify different empirical decisions (like ensembling the logits instead of interpolating the prompts themselves to create a new prompt for the target domain).

**Summary Of The Review:**

Overall, I found the paper well-written, the problem explored is interesting and relevant and the proposed method seems to outperform previous work on a variety of datasets. The empirical analysis is interesting but a couple other design choices / ablations should also be explored (see above). I’m not an expert in NLP so it’s hard for me to judge if the previous work considered covers all the relevant related work and whether the datasets used are the most relevant to study. However, I recommend weak acceptance for the above reasons.

---

> ### Author Response · Authors · 2022-11-16
> **Response to Reviewer eQBR**
>
> We thank the reviewer for their thorough review and detailed suggestions to strengthen our manuscript. We have added experiments and modified our manuscript accordingly. We list the reviewer’s questions/suggestions and our replies below.
>
> &nbsp;&nbsp;&nbsp;&nbsp;&nbsp;&nbsp;&nbsp;&nbsp;&nbsp;_1. Discussions on the inference time of SESoM._
>
> Our reply:  Compared to non-ensemble baselines on the same pre-trained language model, SESoM indeed increases inference time due to the multiple forward passes of source models. Although SESoM requires more forward passes given one target sample, it requires a smaller pre-trained language model (shorter inference time per forward pass) to achieve the same /or even better performance.  SESoM with a small pre-trained model can outperform non-ensemble baselines with a much larger pre-trained model (longer inference time per forward pass).  For example, SESoM with T5-base outperforms ATTEMPT with T5-XL on few-shot classification (please see Table 1). The inference time of SESoM with T5-base is 31.38s while the inference of ATTEMPT with T5-XL takes 38.13s, which shows that the inference time of SESoM with T5-base is shorter. It indicates that SESoM can achieve better performance with a shorter inference time. We have added a more detailed discussion about inference time in _**Appendix B.6**_ and will keep updating it with more statistics.
>
> &nbsp;&nbsp;&nbsp;&nbsp;&nbsp;&nbsp;&nbsp;&nbsp;&nbsp;_2. Results on fewer than 32 shots of training data._
>
> We report SESoM’s results of 8 shots in _**Appendix D.1**_, compared to the best-performing baseline FW-ensemble. SESoM significantly outperforms FW-ensemble in this setting. Moreover, the performance of SESoM with 8-shot training samples is also better than all the baselines with 32-shot training samples.
>
>
> &nbsp;&nbsp;&nbsp;&nbsp;&nbsp;&nbsp;&nbsp;&nbsp;&nbsp;_3. Another ablation: applying the proposed model without first finetuning the prefix of each source model on the target data (that is, directly using the prefix that was trained for each source task, and applying the attention model on logits produced with those prefixes)._
>
> We report results with and without fine-tuning source prompts on target datasets in _**Appendix D.2**_.
> We show that with only 8 and 32 training samples,  SESoM with fine-tuned source models cannot achieve significantly better performance than SESoM without fine-tuned source models. It is likely due to the lack of fine-tuning data.
> However, when the number of few-shot labeled target data increases, SESoM with fine-tuned source models achieve better performance.
>
>
> &nbsp;&nbsp;&nbsp;&nbsp;&nbsp;&nbsp;&nbsp;&nbsp;&nbsp;_4. Analysis on a ‘hard ensembling’ variant: after attention weights are computed, set the argmax position to 1 and all remaining entries to 0s. If this hard variant performs worse than the proposed method, it would show that ensembling is useful compared to selecting the single most appropriate source model._
>
> We add this hard ensembling variant in _**Appendix D.3**_.
> Results show that  SESoM can outperform hard ensembling. As the reviewer suggests, it indicates that our ensemble attention is useful compared to selecting the single most appropriate source model.
>
> &nbsp;&nbsp;&nbsp;&nbsp;&nbsp;&nbsp;&nbsp;&nbsp;&nbsp;_5. Regarding source and target tasks._
>
> We add detailed descriptions of source and target tasks to _**Appendix A**_. Regarding why we choose the current set of source and target tasks, we have followed our most recent and direct baselines (Asai et al., 2022; Gu et al., 2022).
> These tasks cover diverse tasks, domains, and output formats (i.e., span extraction, multiple-choice classification).
>
> &nbsp;&nbsp;&nbsp;&nbsp;&nbsp;&nbsp;&nbsp;&nbsp;&nbsp;_6. In Table 3, why are some of the previous methods and baselines missing?_
>
>
>  In Table 3, we only involve those baselines with the best performance in Table 1.
> Because the original papers of these baselines have proved a better performance compared with the remaining baselines in Table 1 on the full datasets. The purpose of Table 3 is to show how SESoM would perform when we increase the number of shots. We showed that our system not only works for few-shot settings but also works for larger datasets or full datasets.
>
>
> &nbsp;&nbsp;&nbsp;&nbsp;&nbsp;&nbsp;&nbsp;&nbsp;&nbsp;_7. Code release._
>
> We submitted our codes in the original submission.
>
> &nbsp;&nbsp;&nbsp;&nbsp;&nbsp;&nbsp;&nbsp;&nbsp;&nbsp;_8. Other presentation flaws._
>
>
> Regarding Section 3.2.1, 2 lines above Equation 2, we rewrite this sentence to make it clear. The horizontal bar indicates “which is”.  h_x is the final representation of X.
>
> Regarding Algorithm 1 (In the Appendix), we fix the typo and also write a description of the algorithm in _**Appendix B.5**_.
>
> Regarding the location of the discussion about parameter efficiency, we move Parameter efficiency to Section 3.

---

### Official Review · Reviewer_HJCk · 2022-10-24

**Confidence:** 2
**Correctness:** 4
**Technical Novelty And Significance:** 3
**Empirical Novelty And Significance:** 3
**Recommendation:** 6

**Clarity, Quality, Novelty And Reproducibility:**

The paper is well written. The quality is high. To the best of my knowledge, the paper is novel.

**Strength And Weaknesses:**

Strength:

1. The idea to use ensemble is straightforward while effective. It makes the idea simple and easy to implement.
2. The experiment results are good and improve the previous SOTA on most of the tasks considered in this paper.
3. The paper is well written and easy to understand.

Weakness: I am not an expert in knowledge transfer and prompt tuning, thus I do not know the explicit weakness of this paper.

**Summary Of The Paper:**

This paper proposes model ensemble for few-shot prompt tuning in knowledge transfer tasks. The paper use an attention module to do the sample-specific ensemble for different tasks' soft prompt and apply the new ensemble of logits to the new task. Numerical experiments are presented to show the effectiveness of the idea.

**Summary Of The Review:**

Based on my lack of expertise, I think this paper is interesting. I would like to weakly accept this paper while maintaining a low confidence.

---

> ### Author Response · Authors · 2022-11-16
> **Response to Reviewer HJCK**
>
> We would like to thank the reviewer for noting that SESoM is a straightforward, easy to implement, and effective method. We appreciate the reviewer’s viewpoint that the method design being simple is a strength of the method.
>
>
> With this simple and effective architecture, SESoM properly transfers knowledge from different source tasks to a target task in the few-shot prompt-tuning setting.
> We showed that a weighted ensemble of source models’ output given a target sample, can be the most effective knowledge transfer approach for few-shot prompt tuning.
> It is not realized by any previous work, where weighted prompt fusion or prompt re-initialization are still used.
> Therefore, we believe that our work showing that a weighted ensemble of source models’ outputs is actually the current most effective knowledge transfer approach for few-shot prompt tuning, is novel.

---

### Official Review · Reviewer_ATJm · 2022-10-25

**Confidence:** 3
**Clarity, Quality, Novelty And Reproducibility:** The paper is well-written but the ide…
**Correctness:** 2
**Technical Novelty And Significance:** 2
**Empirical Novelty And Significance:** 2
**Recommendation:** 6

**Strength And Weaknesses:**

The main strengths:
1. The paper is well-written and the idea is well-motivated.

2. The experiments are quite comprehensive. The empirical analysis section answers several interesting questions.

3. The empirical study that compares the prediction ensemble with the prompt ensemble is quite interesting and can inspire many related fields.

The main weaknesses:
1. The idea that combining the output of several models using the attention strategy is not novel in deep learning.

2. I can not understand why sample-specific rather than task-specific preference is important for prompt tuning. What if a similar sample exists in a quite different source task? Will the attention model pays more attention to this similar sample resulting in a negative impact on the target task performance since the source task and target task are quite different?

**Summary Of The Paper:**

The paper prose SESoM can adjust the contribution of each model when ensembling the model outputs. SESoM uses an attention module to calculate the attention of each model’s output. The proposed SESoM can achieve SOTA performance in both full-data settings and few-shot settings.

**Summary Of The Review:**

The paper prose SESoM can adjust the contribution of each model when ensembling the model outputs.  The experiments are comprehensive. The idea is not novel.

---

> ### Author Response · Authors · 2022-11-16
> **Response to Reviewer ATJm**
>
> We would like to thank Reviewer ATJm for noting that our idea is well-motivated and that the empirical analysis are comprehensive. We try to address the reviewer’s questions below.
>
>
> &nbsp;&nbsp;&nbsp;&nbsp;&nbsp;&nbsp;&nbsp;&nbsp;&nbsp;_1. Why sample-specific rather than task-specific preference?  If a sample similar to a target sample, exists in a quite different source task, will the attention model pays more attention to this similar sample resulting in a negative impact on the target task performance since the source task and target task are quite different?_
>
>
> We thank the reviewer for raising questions on this corner case.
>
> SESoM generates target-sample-specific preferences for source tasks/models, instead of target-sample-specific preferences for source samples.  Since a source prompt is trained with all training data from the corresponding source task, we expect it to learn the general capabilities required to solve the source task. Therefore, if given a target sample, there exists one source sample very similar to it but with a different label, it shouldn’t lead SESoM to overly rely on this source model/prompt trained with all its training data.
> In the extreme case of this scenario, in which one source task shares the same training data set with the target task, while the labels of them in the source task are the exact opposite of the labels of them in the target task, then this source model/prompt trained with such data is devastating in general for any target sample of the current target task. Then given a target sample, SESoM should assign a very low weight to this source task, since it is trained to capture preferences for general source prompts.
>
> Moreover, while learning target-sample-specific preferences, we have found empirically that SESoM is also able to capture the target task’s general preference for source tasks correctly. The empirical analysis is shown in _**Appendix C**_. If we average the target-sample-specific source-model-weights over all target samples, we can find that SESoM gives higher average attention scores to source tasks that are more similar to the target task.
>
>
> &nbsp;&nbsp;&nbsp;&nbsp;&nbsp;&nbsp;&nbsp;&nbsp;&nbsp;_2. The Novelty of the method._
>
> Although the high-level concept of the ‘weighted ensemble of models’ outputs’ has been used in other domains before, adapting it to transfer knowledge from multiple source prompts for few-shot prompt-tuning is new. In our paper, we try to find a proper knowledge transfer approach from different source tasks to a target task in the few-shot prompt-tuning setting. Previous work has not realized that a weighted ensemble of source models’ output given a target sample, can be the most effective knowledge transfer approach for few-shot prompt tuning. Therefore they still either use weighted prompt fusion or prompt re-initialization.
>
> While in our paper, we not only empirically verified that a weighted ensemble of source models’ output works better than existing methods,  but also propose a new mechanism that tries to measure the relevance of each source task/ prompt for a given target sample, to improve the effectiveness of knowledge transfer through models' outputs ensembling. The simplicity of our attention mechanism allows it to be effective with just a few (8-32) training samples, significantly outperforming existing knowledge transfer approaches in this setting. Therefore, to the best of our knowledge, our adaptation of the idea of 'weighted ensemble of models' to the few-shot prompt-tuning, and our proposed algorithm to execute such adaptation,  are new.

---

### Author Response · Authors · 2022-11-16
**Summary of Changes**

We would like to thank the reviewers for their time spent reading and suggesting improvements to our work. We have made efforts to clarify every point raised by the reviewers in our revised manuscript and have detailed a summary of the changes/clarifications point by point in the text below in addition to providing a more detailed response to each reviewer.

Summary of major changes in our revised manuscript made according to suggestions of reviewers:


* More experiments and empirical analysis: In Appendix D.1, we explored our system with 8 samples of few-shot training labels and showed its better performance than all the baselines. We also showed that with larger samples of few-shot training labels, training each source model with the few-shot target data in prompt-tuning improves the performance of our system in Appendix D.2. Moreover, we added one more variant of our ensemble method in Appendix D.3 to prove that our ensemble attention is useful compared to selecting the single most appropriate source model.
* Additional training and inference details: We added a detailed description of the tasks used in Appendix A. We also elaborated details of the algorithm in Appendix B.5. Moreover, we added the discussion about inference time in Appendix B.6.

---

### Decision · Program_Chairs · 2023-01-20

**Decision:**

Accept: poster

**Justification For Why Not Higher Score:**

The approach is probably overly complicated to see wide adoption or significantly change the field, but is nevertheless sufficiently different from standard practice that it should be accepted.

**Justification For Why Not Lower Score:**

All reviewers recommended acceptance.

**Metareview: Summary, Strengths And Weaknesses:**

This paper proposes a technique for leveraging existing trained soft prompts to attain strong performance in few-shot learning on a new task. Specifically, the technique introduces a module that computes a weighted average of model outputs for different soft prompts. The module is updated on the target few-shot dataset. On average across tasks, the proposed approach significantly outperforms prior methods. Reviewers questioned the novelty, but also appreciated that this approach alleviates many difficulties with few-shot prompt tuning. All reviewers recommended acceptance.

**Note From Pc:**

if the above contains the word "oral" or "spotlight" please see: "oral" presentation means -> notable-top-5% and "spotlight" means -> notable-top-25%. As stated in our emails, we are disassociating presentation type from AC recommendations